# Verapamil and Its Role in Diabetes

Paul Zimmermann [1,2], Felix Aberer [3] , Max L. Eckstein [1], Sandra Haupt [1], Maximilian P. Erlmann [1] and Othmar Moser [1,3,*]

1 Division of Exercise Physiology and Metabolism, Department of Sport Science, University of Bayreuth, 95440 Bayreuth, Germany; paul.zimmermann@arcormail.de (P.Z.); max.eckstein@uni-bayreuth.de (M.L.E.); sandra.haupt@uni-bayreuth.de (S.H.); maximilian.erlmann@uni-bayreuth.de (M.P.E.)
2 Department of Cardiology, Klinikum Bamberg, 96049 Bamberg, Germany
3 Division of Endocrinology and Diabetology, Medical University of Graz, 8010 Graz, Austria; felix.aberer@medunigraz.at
* Correspondence: othmar.moser@uni-bayreuth.de

**Abstract:** Autoimmune pancreatic β-cell loss and destruction play a key role in the pathogenesis and development of type 1 diabetes, with a prospective increased risk for developing micro- and macrovascular complications. In this regard, orally administrated verapamil, a calcium channel antagonist, usually intended for use as an anti-arrhythmic drug, has previously shown potential beneficial effects on β-cell preservation in new-onset type 1 diabetes. Furthermore, observational data suggest a reduced risk of type 2 diabetes development. The underlying pathophysiological mechanisms are not well investigated and remain widely inconclusive. The aim of this narrative review was to detail the role of verapamil in promoting endogenous β-cell function, potentially eligible for early treatment in type 1 diabetes, and to summarize existing evidence on its effect on glycemia in individuals with type 2 diabetes.

**Keywords:** type 1 diabetes; type 2 diabetes; insulin; beta cell preservation; verapamil; thioredoxin-interacting protein (TXNIP)



## 1. Introduction

Approximately 537 million people globally suffer from type 1 (T1D) and type 2 diabetes mellitus (T2D) and prevalence of both is substantially increasing [1,2]. Without sufficient action to address this situation, the number of people suffering from diabetes is predicted to be 643 million in 2030 [2]. The key factor in developing T1D and advanced T2D is the loss or impairment of the insulin-secreting β-cells of the pancreas. In the last 100 years daily insulin injections have been established as the life-saving treatment for most people with T1D and some with T2D. Nevertheless, despite emerging advancements in insulin development and diabetes technology, the majority of people living with diabetes do not achieve individual therapy goals, increasing their risk of acute and late complications [3].

Pancreatic β-cell loss and destruction play a key role in the pathogenesis and development of T1D. In the pancreatic tissue, islets of Langerhans secrete several different hormones, which are responsible for maintenance of glucose homeostasis. Insulin, the only hormone able to lower blood glucose concentration, is secreted by the β-cells, which represent the major cellular component of the pancreatic isles [4]. The primary physiological stimulus for insulin secretion is known to be the increase of circulating glucose concentration. The direct insulin secretion by glucose involves a "triggering" and an "amplifying" pathway. The "triggering" pathway is activated by several biochemical signals, involving the adenosine triphosphate (ATP) generation by glucose metabolism, the closure of ATP-sensitive potassium ($K_{ATP}$) channels resulting in membrane depolarization and consequent activation of voltage-gated calcium channels. The subsequent sharp rise of intracellular calcium levels contributes to the triggered exocytosis of readily releasable pooled insulin

secretory granules by membrane fusion and release to the cell exterior. After the "first phase" of insulin release resulting in a sharp peak, the amplifying pathway provides lower but sustained insulin release for several hours in the "second phase" of insulin secretion. The amplifying pathway is activated in the presence of maximal intracellular $Ca^{2+}$ levels and is largely independent of $K_{ATP}$ driven mechanisms [5].

Residual c-peptide levels representing a consistent and sensitive measure of β-cell function [6] and being detected in many people for years following the diagnosis of T1D, contribute to β-cell responsiveness to hyperglycemia and α-cell responsiveness by reciprocal regulation of glucagon secretion to hypoglycemia for glycemic control in individuals with T1D [7]. Carr et al. demonstrated that detectable c-peptide is associated with an increased time spent in the normal glucose range and with less hyperglycemic episodes, but not with the risk of hypoglycemia in those with newly diagnosed T1D [8]. Preserved c-peptide levels in T1D were associated with a more pronounced counter regulation in response to clamp-induced hypoglycemia [9]. On the one hand, regular physical exercise contributes to β-cell preservation, improved insulin sensitivity and less requirements of exogenous insulin administration [10]; on the other hand, in T1D subjects undertaking high levels of physical exercise, the honeymoon period, which is defined by an absence of insulin requirements early after onset of diabetes, is five times longer compared to matched sedentary controls [11]. Next to physical activity, conscious macronutrient intake, such as gluten deprivation or reduced consumption of refined grains may have beneficial effects on β-cell preservation in people affected by new-onset T1D [12,13].

Individuals with T1D are exposed to an increased risk for developing micro- and macrovascular complications, which are associated with episodes of dysglycemia. In this context, residual β-cell secretion, evaluated by measuring fasting c-peptide levels, has been shown to be prospectively associated with reduced incidence of microvascular complications in T1D [14]. Even modestly detectable β-cell levels correlated with a reduced incidence of diabetes-related complications, such as retinopathy and nephropathy [15].

By the fact that autoimmune-mediated β-cell destruction is unavoidably progressing, sooner or later, complex insulin therapy is required for the lifetime. The time from T1D diagnosis to complete lack of measurable insulin (c-peptide) is highly individual as shown by Davies et al., who demonstrated that after 6–9 years of diabetes diagnosis, insulin remained detectable in 60% of individuals, while after 10–20 years of diabetes duration only 35% of the individuals remained c-peptide positive, as defined by detectable fasting c-peptide $\geq$ 0.017 nmol/L and non-fasting c-peptide $\geq$ 0.2 nmol/L [16]. Recent research on T1D enables us to refine our understanding in pathogenesis and subsequent development of insulin deficiency in T1D and potentially establish novel prevention and therapy strategies [17]. The impairment of β-cells leads to long-term immune-mediated destruction, low insulin secretory capacity and autoantigen presentation [17]. However, up to now, evidence on effective therapies to delay or halt this process is largely lacking [18].

For these reasons, β-cell rescue and preservation strategies are hot topics on current and future therapeutic strategies in T1D. Exploring beneficial actions for the treatment of T1D, clinical data are suggesting positive effects for peptides or medication reducing the β-cell stress, such as verapamil [17,19,20]. Verapamil was the first non-dihydropyridine calcium channel blocker (CCB) that was approved by the Food and Drug Administration (FDA) in 1981 for clinical use [21]. In several clinical implications, such as cardiac arrhythmias or combination treatment of hypertension, it has proven efficacy in everyday clinical practice due to its good safety profile and pharmacodynamic properties [22,23].

Next to the previously described impact of preserved endogenous insulin secretion, measured by c-peptide levels in T1D individuals, the role of c-peptide is not well defined in T2D, a disease that is considered to be associated with insulin resistance and a reduced β-cell function [24]. Therefore, preserving β-cells function is one of the principle aims in the treatment of T2D to delay the natural course of the disease, necessitating the introduction of insulin therapy in the majority of patients [14]. In T2D patients, regular moderate physical activity and physical health represent well accepted key factors next to regular orally

administrated antidiabetic medication and finally additional exogenous insulin application combined with conscious macronutrient intake in order to prospectively hold back the progress of β-cell decline and insulin resistance. In this matter, several clinical observational studies in general reported about decreased risk of new onset diabetes and lower fasting blood glucose levels in diabetes patients receiving orally administrated verapamil [25–27]. Dietary factors are estimated to contribute to maintaining insulin secretion and sensitivity by reduced consumption of refined grains and meat products in T2D [13]. Some prospective studies reported a positive association between residual insulin secretion in T2D patients and less microvascular complications [28], but up to now, to our knowledge, no data regarding the association between residual insulin secretion and major outcomes, such as all-cause mortality and mortality due to cardiovascular diseases, are available in T2D patients [29].

In this regard, we review the role of orally administrated verapamil in diabetes positively influencing β-cell function and glycemic control as well as its potential properties to prevent diabetes development.

## 2. Method Section

*Scientific Research*

We selected relevant scientific research published from October 1984 until May 2022 by searching PubMed. Potentially eligible studies were considered to be included in our narrative review after searching by combined-term medial subject headings and keywords, such as type 1 diabetes (T1D), type 2 diabetes (T2D), insulin secretion, β-cell preservation, verapamil, and Thioredoxin-interacting protein (TXNIP). After completing the search, 69 papers and one web source were included to detail the systemic and cellular effects of orally administrated CCB verapamil in T1D and T2D subjects.

## 3. Insulin Secretion in Pancreatic β-Cells and the Role of TXNIP—Influence of Verapamil and Clinical Implications

*3.1. The Role of Pancreatic β-Cells in T1D and T2D*

T1D is an autoimmune-mediated disease characterized by progressive destruction of the pancreatic β-cells resulting in long term lack of the hormone insulin [30]. Pancreatic β-cells play a pivotal role in the synthetization and secretion of insulin, as the body's solo source [31]. In this regard, insulin represents the main player for promotion and maintenance of metabolism [32]. In the scientific community it is an accepted fact that diabetes is associated with a reduction in β-cell mass and to date there is no approved drug treatment that targets damage to these cells [33,34]. Pancreatic β-cells have, as reported in several studies, a weak antioxidant capacity and are very sensitive to oxidative stress interactions occurring within the cells [31,33]. Although several trials have studied the mechanisms of β-cell loss in the different types of diabetes, there is less information referring to the residual β-cells in autoimmune T1D [35].

Different mechanisms are postulated for β-cell failure as demonstrated especially for T2D individuals [36]. β-cells in T2D patients are reported to be secretory-functionally inactive for decades and their potential and preserving might contribute to new therapeutical approaches [35]. A heated discussion is ongoing whether a β-cell reduction occurs in every person with T2D. Some researchers argue that the β-cell mass in T2D patients stays normal and remarks the functional abnormality of insulin secretion as the main problem of hyperglycemia. On the other hand, the scientific community discusses the reduction of the absolute β-cell mass, which is far more difficult to restore [34]. In this context, oxidative stress might inactivate key islet transcription factors, producing "stunned" β-cells, not responding to glucose [36–38]. Although it is difficult to measure the β-cell mass in vivo, there is a proposed positive correlation between high mass and high insulin sensitivity and secretion [34,39,40].

### 3.2. Thioredoxin-Interacting Protein (TXNIP) and Its Regulation of Pancreatic β-Cells

Thioredoxin-interacting protein (TXNIP) is an attractive aspect to focus on, as it has been suggested to be a major factor in the regulation of pancreatic β-cell dysfunction and death, representing key processes in the pathogenesis of T1D and T2D [19,41]. Therefore, TXNIP represents a very promising future target in the therapy for diabetes based on basic, preclinical and retrospective epidemiological analyses [41,42]. TXNIP inhibits thioredoxin (TRX) as a part of the intracellular antioxidant system, which manages different mechanisms in the β-cells, mainly the reduction of the antioxidant capacity and subsequent oxidative stress and apoptosis in the β-cells, resulting in reduced insulin production capacity (Figure 1) [43,44].

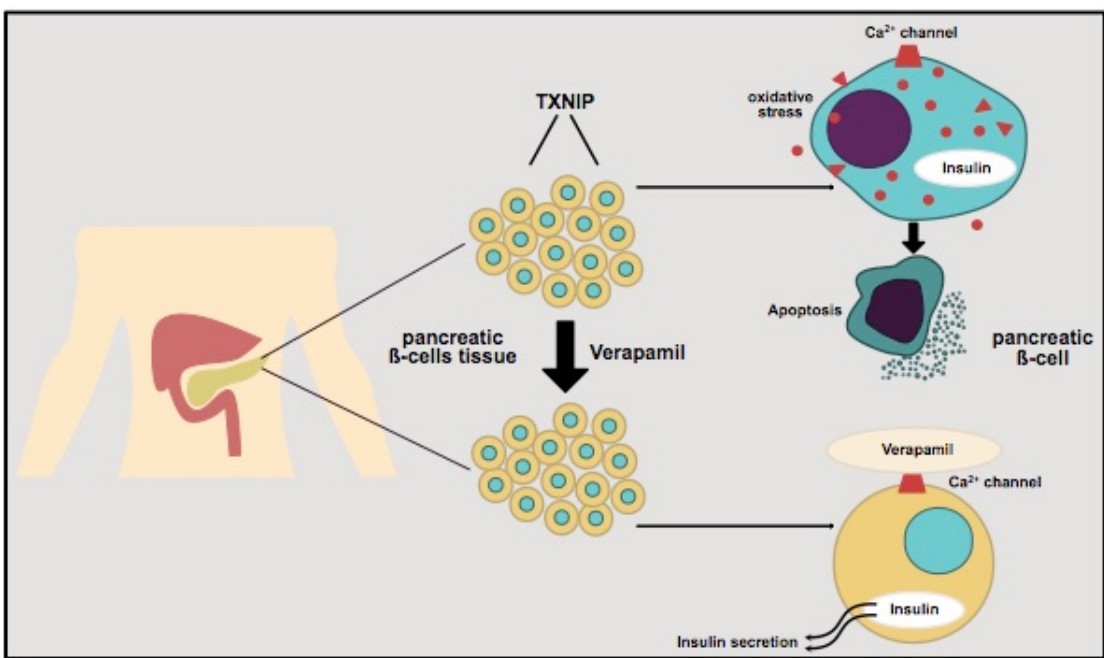

**Figure 1.** The role of pancreatic β-cells, oxidative stress and insulin secretion in T1D, [31]. Abbreviations: TXNIP, Thioredoxin interacting protein; $Ca^{2+}$, Calcium.

In detail, TXNIP regulates the glucose homeostasis as a signal complex, the TRX/TXNIP signal complex. This redoxisome represents the basis of TXNIP regulation as redox response. TXNIP has been shown to bind NOD-like receptor protein 3 (NLRP3) and activate the inflammasome [41]. TXNIP as a member of the ancestral α- Arrestin family binds to the Itchy E3 Ubiquitin Protein Ligase (ITCH) and enables the ubiquitination of the substrates. TXNIP in general is transcriptionally regulated by nuclear receptors (NR), such as glucocorticoid receptor (GR), vitamin D receptor (VDR), farnesoid X receptor (FXR) and peroxisome-proliferator activated receptor (PPARs) in a cell-specific manner [41]. These signal complex regulators are involved in the physiological regulation functions of TXNIP, for example in the regulation of glucose homeostasis, as pictured in Figure 2.

TXNIP has been shown to be activated by hyperglycemia and to be increased in diabetes, whereas TXNIP deletion seems to be associated with non-diabetes occurrence in general. In detail, TXNIP is one of the genes that is highly upregulated by hyperglycemia in murine and human β-cells. Therefore, in the case of β-cells the glucose sensor carbohydrate-response element-binding protein (ChREBP) directly binds to the promoter region of TXNIP and increases gene expression [41,45,46]. Furthermore, TXNIP inhibition has been proven for promoting insulin production and glucagon-like peptide 1 signaling via the microRBA regulation [42]. On the other hand, the glucose responsiveness of TXNIP is linked to the notable functions of induction of apoptosis as a reaction to hyperglycemic episodes [41,45,47]. This stress-induced upregulation of TXNIP can be noticed in the

pancreatic islets during progression of diabetes in humans and mice [48,49]. Varying factors, such as glucocorticoids, lipids, inflammation/cytokines and oxidative stress, which influence and stimulate the TXNIP induction, are described in previous research [50–53].

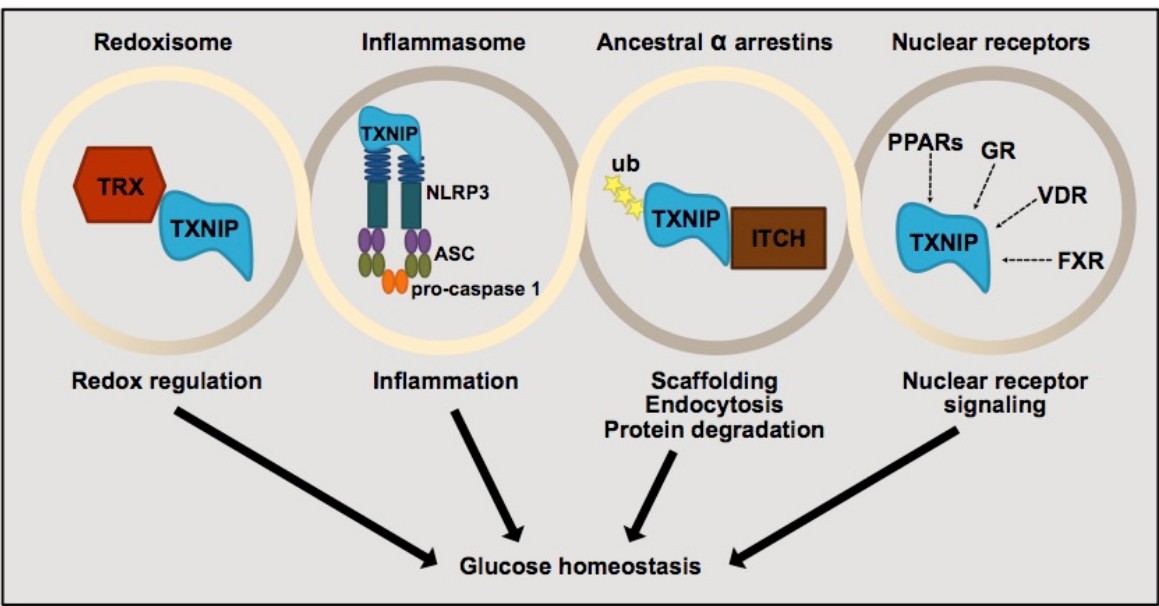

**Figure 2.** TXNIP signal complex regulating glucose homeostasis, [41]. Abbreviations: TRX, Thioredoxin; TXNIP, Thioredoxin interacting protein; NLRP3, NOD-like receptor protein 3; ITCH, Itchy E3 Ubiquitin Protein Ligase; PPARs, peroxisome-proliferator activated receptors; GR, glucocorticoid receptor; VDR, vitamin D receptor; FXR, farnesoid receptor.

In this context, there has been shown some scientific evidence that pancreatic β-cells as well as skeletal myocytes share common mechanisms of fuel sensing in order to cooperate and maintain glucose homeostasis in the whole-body system. Therefore, TXNIP has been recently shown to play a key role as a diabetogenic culprit disrupting the following both processes—on the one hand by activating the pancreatic isles by mobilizing insulin-containing vesicles and on the other hand by modulating the translocation of resident glucose transporters in the peripheries of the muscles [48,54]. The thioredoxin system plays an important role at a nodal point linking pathways of redox regulation, energy metabolism, antioxidant defense, and in the end cell growth and survival [44]. Hypoglycemic agents, carbohydrate-response-element-binding protein and cytosolic calcium levels regulate the β-cell TXNIP expression, and these different aspects contribute to regulation of whole-body glucose maintenance [42]. This vicious cycle may contribute to TXNIP triggered β-cell failure and overt diabetes [44].

Next to the mentioned TXNIP interactions, TXNIP is an α-Arrestin that acts as an adaptor for glucose transporter 1 (GLUT1), which plays upregulated—as a major glucose facilitator—an important role in the development of metabolic diseases, such as diabetes. TXNIP interacts with GLUT1 lipid nanodiscs in a 1:1 ratio and regulates the glucose uptake in response to intracellular as well as extracellular signals. TXNIP-GLUT1 interaction depends on TXNIP interaction with phosphatidylinositol 4,5-bisphosphate PI(4,5)P2 or PIP2 and TIXNP does not interact with GLUT5 [55].

In summary, the TRX/TXNIP signal complex has been shown to play an important role in the redox-related signal transduction in many different types of cells in various tissues. Additionally, TXNIP has several cellular functions, which largely rely on their scaffolding function as a member of the α-Arrestin family [41]. By both functions, i.e., the redox dependent and independent, TXNIP has emerged as master regulator for glucose homeostasis. Targeting TXNIP in diabetes seems to play an important role in the whole-

body glucose metabolism regulation influenced by variable factors and circumstances and in the future might inaugurate new therapeutical potential in diabetes therapy.

### 3.3. Verapamil and Its Impact on Diabetes

The non-dihydropyridine CCB verapamil and its role in clinical routine as cardiac antiarrhythmic therapy and a blood-pressure-lowering drug was approved by the FDA in 1981 due to its advantageous pharmacodynamics for the treatment of angina, hypertension, supraventricular tachycardia and atrial fibrillation [21,22]. In recent years it has been considered as a promising novel approach in the therapy of TD1 and T2D [21]. The cardiac side effects and antidiabetic efficacy of R-form verapamil enantiomer (R-Vera) and S-form verapamil enantiomer (S-Vera) were evaluated in mouse models and R-Vera seems to represent an effective option in diabetes treatment by downregulating TXNIP and reducing β-cell apoptosis with an established safety profile and only weak adverse cardiac effects, such as negative inotropy [21]. While the rise of intracellular calcium concentration is known in general as the main trigger of exocytosis and subsequent insulin secretion, verapamil reduces by blocking calcium channels the intracellular calcium concentration and prevents long-term β-cell impairment, which is partly caused by chronic increased intracellular $Ca^{2+}$ levels. This preventive mechanism contributes to preserved β-cell function by TXNIP downregulation, ameliorating less apoptosis in pancreatic β-cells and helping to preserve continuously endogenous insulin levels during glucose metabolism regulation [21].

In general, the three different subtypes of calcium channels, i.e., $Ca_V$ 3.1, -3.2 and -3.3, are distributed over the whole body and have defined roles in cardiac regulation, vasculature tone regulation and the activation of the nervous system. The main effect of verapamil results in blocking of both L-Type and T-type channels with higher affinity for depolarized channels than for resting channels. The highest affinity, up to ten times higher, of verapamil is reported in depolarized L-type channels than in the resting channels [22]. The phenylalkylamine Br-verapamil binds in the central cavity of the pore on the intracellular side of the selectivity filter—blocking the ion-conducting pathway—and structure-based mutations of key amino-acid residues confirm the verapamil binding on both sides, as reported by Tang et al. [56]. These specific positive effects could be verified in several studies, as shown in mouse models resulting in improved β-cell survival and function, enhanced insulin secretion and reduced diabetes rate [19]. Next to these findings, several clinical observational studies, such as International Verapamil SR/Trandolapril (INVEST) and the Reasons for Geographic and Racial Differences in Stroke (REGARDS) study, confirmed decreased risk for newly diagnosed diabetes and lower fasting blood glucose levels in response to regular oral verapamil intake [25–27].

### 3.3.1. Verapamil Administration and β-Cell Mass in Mouse Model

In this regard TXNIP was identified as a target to halt the functional β-cell mass loss as described by Xu et al. in a mouse model [19]. Hyperglycemia and diabetes induce an upregulation of β-cell TXNIP expression, and TXNIP overexpression causes β-cell apoptosis. Although it has previously been shown that TXNIP is strongly dependent and induced by glucose, different proinflammatory cytokines, such as tumor necrosis factor α (TNF α), interleukin-1 β (IL-1 β) and interferon γ (IFNγ) each have distinct and partly opposing mechanisms and pathways on β-cell TXNIP expression [19,50].

Xu et al. could reveal positive effects due to inhibition of TXNIP expression, enhanced endogenous insulin levels as well as improved glucose homeostasis and sensitivity in the mouse model. These positive effects of TXNIP repression by orally administrated verapamil in a mouse model seem to be conditional by reduction of intracellular calcium levels, inhibition of calcineurin signaling and nuclear exclusion and decreased binding of carbohydrate response element-binding protein to the E-box repeat in the TXNIP promoter [19].

For the first time it was highlighted that oral medication of the CCB verapamil could effectively inhibit proapoptotic β-cell TXNIP expression, improve β-cell survival and

function with weak adverse cardiac effects, and could represent a new therapeutic approach for the prevention and therapy of diabetes.

### 3.3.2. Clinical Implications in Type 1 Diabetes (T1D)

To translate these positive findings reported in a mouse model [19,50] into humans, Ovalle et al. performed a trial in order to assess the efficacy and safety of using oral verapamil in subjects with recent onset T1D in order to downregulate TXNIP and enhance the patients' endogenous β-cells mass and insulin production [18]. Therefore, in a double-blind, placebo-controlled Phase 2 trial, 32 participants were randomized to assess the efficacy and safety of orally administrated verapamil in subjects with recent onset T1D in order to downregulate TXNIP, and to evaluate the maintenance of endogenous β-cell mass and insulin production. Furthermore, 26 participants were randomized to the two treatment groups, i.e., placebo control group versus oral medication with verapamil for 12 months. The initial dose of verapamil was 120 mg daily and was advanced to a maximum dose of 360 mg daily, if tolerated. The primary outcome measures assessed the functional β-cell mass by the area under the curve (AUC) from a two-hour mixed meal stimulated c-peptide after 12 months. As secondary outcome measures the changes from baseline in exogenous insulin requirements both within 12 weeks and 12 months, hypoglycemic events as well as the HbA1c values within 12 weeks and 12 months were defined. An improved endogenous β-cell activity, lower exogenous insulin requirements and lower hypoglycemic episodes were demonstrated in the verapamil group for at least 24 months and lost upon discontinuation [18,57]. These positive findings were shown and consistent to the previous results in preclinical diabetic mouse model studies and in isolated human islets [18,50]. Evaluating secondary endpoints, as the total daily dose of insulin (TDDI) to maintain glycemic control, a significant treatment difference of −43% in the verapamil group compared to the placebo group could be revealed within the first follow-up year, as well as non-significant lower HbA1c levels ($p$ = 0.083) in the verapamil group. Moreover, an improved glycemic control with significant less hypoglycemic episodes in the verapamil group ($p$ = 0.0387) as well as more time within the target range of 3.9–10.0 mmol/L assessed by a continuous glucose monitoring (CGM) system were reported within the verapamil group. Verapamil treatment did not affect fastening glucagon levels and no severe adverse events occurred in the verapamil group causing treatment discontinuation. These positive effects, especially the comparable glucagon levels in both groups, might be assumed by an improved insulin sensitivity due to verapamil administration resulting in an overall better glucose control. These aspects might contribute to lower exogenous insulin requirements, which in turn could reduce the hypoglycemic episodes [18]. These different mechanisms might result in an overall improved glucose control and stable glucagon levels in both groups might serve as a positive feedback control mechanism. Importantly, verapamil administration did not cause any severe episodes of hypotension, heart rate abnormalities or electrocardiogram (ECG) alterations. These results emphasize the potential translational implications and its impact on clinical care and encourage the scientific community for larger follow-up trials in order to develop novel therapeutical approaches [18,19].

The clinical implications of TXNIP targeting in T1D subjects seem to preserve additional therapeutic opportunities to decrease long term micro- and macrovascular complications, such as diabetic vascular dysfunction, diabetic retinopathy as well as diabetic nephropathy, and to decrease the rate of diabetes-related morbidity and mortality [58,59]. These positive effects are based on the established mode of verapamil, i.e., the blockade of L-type calcium channels resulting in a decrease of intracellular calcium level followed by an inhibition of TXNIP transcription [19]. In this context, tissue with a high expression level of L-type calcium channels, as the heart or the β-cells, consequently benefits from the TXNIP inhibition and positive effects have been shown in diabetic heart disease [60,61].

A recently published exploratory study of Xu et al. assessed the potential systemic changes in response to verapamil treatment by global proteomics analyzed by using liquid chromatography-tandem mass spectrometry (LC-MS) and revealed positive systemic and

cellular effects of orally administered verapamil in TD1 subjects [57]. The initial trial was registered by clincicaltrials.gov (NCT02372253 2/20/2015) and previous research was published by Ovalle et al. 2018 [18]. The participants were randomized to oral verapamil (360 mg sustained-release daily) or placebo.

In this study, focusing on continuous use of verapamil, several positive effects, such as delayed T1D progression, promotion of endogenous β-cell function and consecutive lowered insulin requirements by continuous verapamil application were revealed. These positive effects were sustained for at least two years by regular application and were lost upon discontinuation. No further follow up after two years was performed in this exploratory trial. Therefore, the current studies point out crucial mechanistic and clinically beneficial effects of administered verapamil in T1D patients [57]. These positive effects of orally administered verapamil might be assumed by TXNIP inhibition causing β-cell protective and anti-diabetic effects. Analyzing chromogranin A (CHGA) serum levels as a potential therapeutic marker before and after treatment revealed a positive correlation with loss of β-cell function, reflected changes in verapamil treatment and discontinuation and persisted over a follow up time of at least two years. In summary, the results of this exploratory study suggested that a continuous orally administered verapamil treatment in T1D individuals may lower insulin requirements and decelerate disease progression for at least two years after diagnosis. These positive effects are associated with normalization of CHGA levels, and anti-oxidative effects, and immunomodulatory gene expression profile in pancreatic isles. The complex interaction is pictured in Figure 3. [57]. All these changes might contribute to the overall beneficial effects of verapamil use.

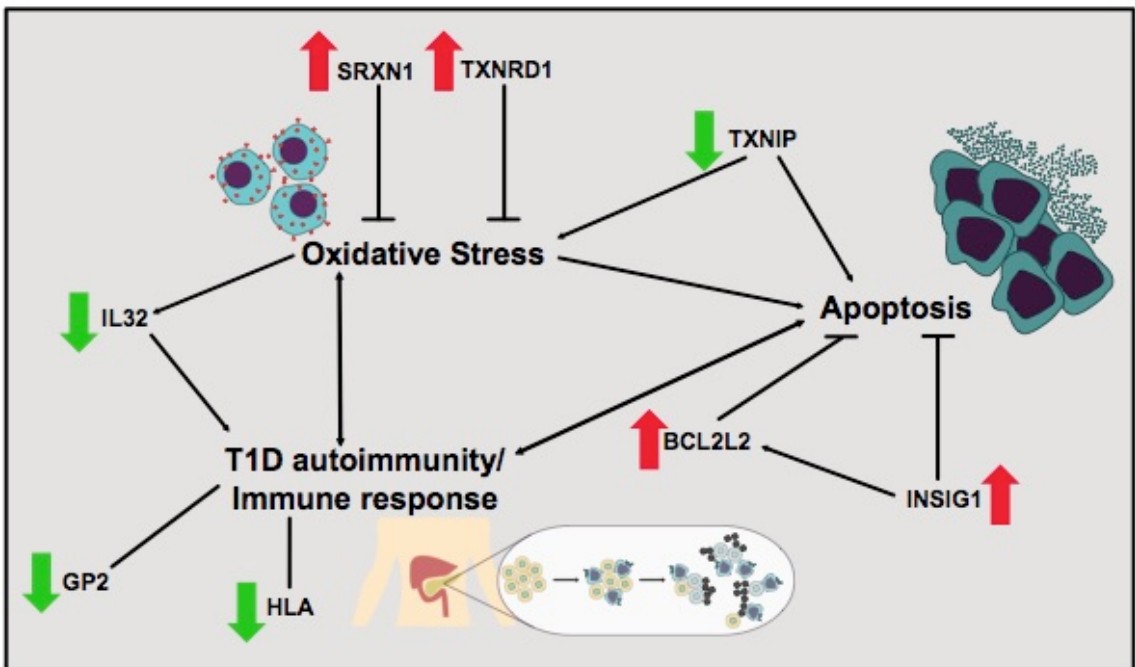

**Figure 3.** Systemic and cellular effects of verapamil treatment in subjects with type 1 diabetes, [57]. Abbreviations: TXNIP, Thioredoxin interacting protein; IL32, interleukin 32; BCL2L2, Bcl-2-like protein2; GP2, glycoprotein2; INSIG1, insulin-induced gene1; HLA, human leucocyte antigen; TXNRD1, Thioredoxin reductase; SRXN1, sulfiredoxin reductase; red arrow, represents upregulation by verapamil; green arrow, represents downregulation by verapamil.

These reported beneficial findings have to be confirmed in larger studies and might improve diabetes control in subjects with T1D in the future [18,57].

### 3.3.3. Clinical Implications in Type 2 Diabetes (T2D)

In a retrospective population-based cohort study from the Taiwan's National Health Insurance Research Database, regular oral verapamil use was associated with a decreased incidence of T2D in patients with unknown history of diabetes in comparison to a matched group of patients treated with other CCB with an adjusted hazard ratio 0.80 [6]. These positive findings are supported by the observational data analyses from the International Verapamil SR/Trandolapril (INVEST) studies, which revealed a lower risk for developing diabetes as well as the data derived from the study using the Reasons for Geographic and Racial Differences in Stroke (REGARDS) cohort, where lower fasting blood glucose levels were shown in patients using verapamil compared to subjects with diabetes without CCB [25–27]. The results of both mentioned observational studies highlight the positive effects of orally administrated verapamil as a potentially preventive agent in T2D development. Next to these preventive aspects, positive results regarding the inhibition of gluconeogenesis are reported in T2D patients, which contributes to improved glucose homeostasis in T2D individuals [62].

Next to the reported studies, which have shown a lower incidence of T2D in verapamil-treated subjects, Malayeri et al. could reveal positive effects in T2D subjects in a randomized, double-blind, placebo-controlled trial [33]. In this study, verapamil administration showed a better glycemic control by means of decrease of HbA1c, decrease of TXNIP expression and increased glucagon-like peptide-1 receptor (GLP1R) mRNA providing increasing β-cell survival [33]. Additional findings by Carbovale et al. revealed significantly lowered plasma glucose levels in verapamil-treated subjects with T2D [63].

On this account, verapamil may serve as an effective oral adjunct therapy in combination with oral antidiabetic drugs in T2D patients in the future as it is safe, improves glycemic control, and might preserve β-cells function as demonstrated in T1D and T2D mouse models [21].

These previously described positive effects of orally administrated verapamil, based on retrospective population-based and observational data analyses [6,25–27] as well as the presented data of a randomized, double-blind, placebo-controlled trial [33], could elucidate the positive effects of TXNIP regulation on glucose metabolism. Additionally, positive findings were revealed by Hong et al. in mouse models of T2D, who demonstrated for the first time new mechanistic insights and novel links between TXNIP and proinflammatory cytokines and microRNA signaling [50]. Furthermore, latest research results by Wu et al. revealed positive effects of verapamil use in type 2 diabetic rats on bone mass, microstructure as well as macro- and nano mechanical properties of the femur [64]. Taken together these several positive effects emphasize the important role of TXNIP and its effects on the pancreatic β-cell and TXNIP expression in T2D and underline through various systemic and cellular effects its potential as an adjunctive therapeutic approach.

## 4. Discussion

Since loss of functional β-cell mass is one of the key aspects of diabetes in general, different therapeutical approaches have been established in past decades in order to halt this process [20]. Chronic increased intracellular $Ca^{2+}$ levels seem to contribute to impaired β-cell function and are associated with long term β-cell impairment. In this regard, excitotoxicity or overnutrition and the combination of both stresses seem to play an important role, as they might cause alterations in the β-cells transcriptome, mitochondrial energy metabolism, fatty acid β-oxidation, and mitochondrial biogenesis [65].

Next to the current physical activity recommendations of 150 min of moderate-intensity aerobic exercise per week resulting in optimized glycemic control in individuals with diabetes [66,67], additional early oral verapamil usage has been reported to improve insulin-stimulated glucose transport in skeletal muscle, resulting in optimized glycemic control and improved insulin sensitivity.

Next to the mentioned positive effects of orally administrated verapamil on the β-cell preservation and the improved glycemic control [31,48], several overall beneficial effects

observed with verapamil have been illustrated [57]. In summary, in our opinion these far reaching cellular and systemic regulatory effects seem to contribute to the positive assessment of verapamil, referring to its impact on diabetes. Next to the regulating effects on the thioredoxin system [57], in individuals with diabetes, who are predisposed to micro- and macrovascular complications during their lifetime, the management of autoimmune-related injury has to be focused. Verapamil promotes by regulation of the thioredoxin system several antioxidative, anti-apoptotic and immunomodulatory interactions in the human pancreatic islets [57]. Current scientific evidence suggests that TXNIP-targeting therapeutics, such as verapamil, seem to play an important role as central regulators of whole-body glucose homeostasis [41]; nevertheless, the basic molecular mechanisms of how TXNIP interacts with other proteins in different cellular tissues is not fully understood. In this context, up to now the interaction between TXNIP and glucagon is not completely understood, but Thielen et al. could identify a novel orally substituted quinazoline sulfonamide, SRI-37330, with an excellent safety profile and inhibition of TXNIP in human islets, inhibition of glucagon function and secretion, lowering hepatic glucose production and strong anti-diabetic effects in a mouse model of T1D [68]. These reported findings on SRI-37330 are consistent with previous observations on TXNIP targeting by blockage of the L-type calcium channels with verapamil. These positive effects for verapamil have been shown in mouse models [19,43], in a randomized controlled trial in individuals with T1D [18], as well as the association with reduced incidence of newly diagnosed T2D [6,25,26,42] and better overall glycemic control in subjects with diabetes [27]. By the lack of validated clinical approaches for detecting insulitis and β-cell decline in T1D preclinical models to diagnose eventual diabetes and to monitor the efficacy of therapeutical interventions, ultrasound imaging of the pancreatic perfusion dynamics revealed delayed diabetes development by orally administrated verapamil [69]. These therapeutical strategies might provide a deployable future predictive marker for therapeutic prevention in asymptomatic T1D individuals [69].

Nevertheless, verapamil is a blood pressure medication and an anti-arrhythmic drug and its TXNIP capacity is linked to its function as L-type calcium channel blocker [68]. Therefore, in our opinion the daily administrated verapamil has to be limited to certain patient populations, especially those who tend to hypotension and left ventricular systolic dysfunction, suffer from hepatopathy or might be predisposed for potential polypharmacy drug interactions. These side effects might prohibit its regular clinical prescription.

Other important points that have to be mentioned are the lack of data referring to the long-term application of verapamil, specifically in its indication as a diabetes-modifying drug. The present exploratory studies reveal some far-reaching systemic and cellular effects of verapamil treatment in the context of T1D [57]. Next to the described preservation of β-cell function in the pancreatic tissue, unappreciated positive connections between immune system, regulation of proinflammatory cytokines, lowering of CHGA in response to verapamil use were revealed and might help to dampen the associated autoimmune processes in T1D [57]. In our opinion, these interesting aspects contribute to the positive overall effects of verapamil in diabetes. Nevertheless, the current scientific studies were limited to small numbers of subjects. In this context, the VER-A-T1D trial (VER-A-T1D; NCT04545151) as a multicenter, randomized, double-blind, placebo-controlled study will evaluate the effect of orally administered verapamil on the preservation of β-cell function as measured by stimulated c-peptide levels after 12 months. Furthermore, another multi-national trial investigates the use of verapamil in children and adolescents with newly diagnosed T1D to assess hybrid closed loop therapy and verapamil for β-cell preservation in new onset T1D (CLVer; NCT04233034), which was initiated in July 2020 and will be completed in September 2022. Nevertheless, the outcomes of both initiated studies and the presented scientific research in general are limited to a small number of participants and a short follow-up time with a lack of long-term follow-up results. Future research and longer follow-up periods will be of great interest for the scientific community, such as safety profile and side effects, as well as their daily practicability regarding the regular

continuous verapamil use as a new innovative therapeutical approach. In the end the previously described positive effects of oral adjunct verapamil administration in subjects with T1D have to be confirmed in larger studies.

## 5. Conclusions

In conclusion, daily orally administered CCB verapamil added early to standard therapy in diabetes, mainly T1D, might contribute to establishing an effective adjuvant T1D therapy. Inhibition of β-cells TXNIP expression seems to represent a new therapeutical approach for the future prevention and therapy of diabetes, while preserving and promoting the person's own endogenous β-cell function as well as optimizing overall glucose control by reducing exogenous insulin requirements and reducing hypoglycemic risk. Next to the mediated β-cell preservation, far-reaching positive systemic and cellular effects by daily orally administered verapamil use seem to dampen the associated autoimmune processes in T1D. In patients with no history of diabetes mellitus, a decreased incidence of T2D could be revealed in observational data analyses compared to the usage of other CCB. This additional safe and effective novel approach might provide an adjunctive therapeutical treatment option in the future management of diabetes mellitus and has to be confirmed in further clinical investigation in larger patient cohorts.

**Author Contributions:** Conceptualization, P.Z. and F.A.; methodology, O.M.; software, P.Z.; validation, O.M.; formal analysis, O.M.; investigation, P.Z.; writing—original draft preparation, P.Z.; writing—review and editing, F.A., M.L.E., S.H., M.P.E. and O.M.; supervision, F.A.; project administration, O.M. All authors have read and agreed to the published version of the manuscript.

**Funding:** Funded by the Deutsche Forschungsgemeinschaft (DFG, German Research Foundation)—491183248.

**Institutional Review Board Statement:** Not applicable.

**Informed Consent Statement:** Not applicable.

**Data Availability Statement:** Not applicable.

**Conflicts of Interest:** M.L.E. received a KESS2/European Social Fund scholarship and travel grants from Novo Nordisk A/S and Sanofi-Aventis. F.A. received speaker honoraria from Eli Lilly, Merck Sharp & Dome, Boehringer Ingelheim, AstraZeneca and Amgen. O.M. has received lecture fees from Medtronic, Sanofi-Aventis travel grants from Novo Nordisk A/S, Novo Nordisk AT, Novo Nordisk UK, Medtronic AT, Sanofi-Aventis, research grants from Sêr Cymru II COFUND fellowship/European Union, Novo Nordisk A/S, Sanofi-Aventis, Dexcom Inc., and Novo Nordisk AT as well as material funding from Abbott Diabetes Care. P.Z. received speaker honoraria from Bayer, Daiichi Sankyō, Amarin Germany GmbH, and AstraZeneca GmbH. All other authors have no conflict of interest relevant to the article to declare.

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
