# Peer review of "Verapamil and Its Role in Diabetes"

_diabetology, doi:10.3390/diabetology3030030_

Round 1
Reviewer 1 Report
In the article titled Verapamil and its role in diabetes, Zimmermann et al. describe the potential role of verapamil in the treatment of type 1 diabetes mellitus. The article is well written, and the authors cite the most relevant and recent work in the field. Below I have listed a few of my inquiries and comments for the article.
1. Line 24: The authors should include the most up-to-date data on the number of diabetics in the world – IDF Diabetes atlas 2021
2. Line 200-203: this has already been mentioned in lines 86-80
3. Line 237-240: In mouse models, verapamil seems to reduce intracellular calcium concentrations, the rise of which is the main trigger of exocytosis and hence insulin secretion. Could authors in this perspective explain the positive effect of verapamil on insulin secretion?
Author Response
Dear Editor,
Dear Reviewers,
Thank you very much for reviewing our manuscript entitled “Verapamil and its role in diabetes”. Please find below a point-to-point response to the specific comments.
Reviewer 1:
Dear Reviewer,
we highly appreciate the willingness to review our manuscript and also express our thanks for the comments and positive feedback for the authors.
In the article titled Verapamil and its role in diabetes, Zimmermann et al. describe the potential role of verapamil in the treatment of type 1 diabetes mellitus. The article is well written, and the authors cite the most relevant and recent work in the field. Below I have listed a few of my inquiries and comments for the article.
- Line 34: The authors should include the most up-to-date data on the number of diabetics in the world – IDF Diabetes atlas 2021
Thank you very much for this important comment. In our revised manuscript we included the most up-to-date data on this scientific topic and the revised the manuscript was amended as following, on page 2, line 35 to 38:
“Approximately 537 million people globally suffer from type 1 (T1D) and type 2 diabetes mellitus (T2D) and prevalence of both is substantially increasing. Without sufficient action to address this situation, the number of people suffering from diabetes is predicted to be 643 million in 2030.”
- Line 200-203: this has already been mentioned in lines 86-80
We would like to thank the reviewer for taking the time to read our manuscript thoroughly and addressing this important point to us. We are of the same opinion that the manuscript has to be modified. Therefore we improved the text of our revised version of the manuscript and amended our manuscript as following on page 3, line 102 to 109:
“Recent research on T1D enables us to refine our understanding in pathogenesis and subsequent development of insulin deficiency in T1D and might establish novel prevention and therapy strategies. The impairment of ß-cells leads to long term immune mediated destruction, low insulin secretory capacity and autoantigen presentation. But up to now, evidence on effective therapies to delay or halt this process is largely lacking.“
- Line 237-240: In mouse models, verapamil seems to reduce intracellular calcium concentrations, the rise of which is the main trigger of exocytosis and hence insulin secretion. Could authors in this perspective explain the positive effect of verapamil on insulin secretion?
We would like to thank the reviewer for taking the time to read our manuscript thoroughly and addressing this important interesting aspect of pathophysiology to us. The reviewer is absolutely right that this point should be addressed in the revised version of our manuscript. Therefore, the manuscript was amended as following on page 8, line 390 to page 9, line 414:
“The cardiac side effects and antidiabetic efficacy of R-form verapamil enantiomer (R-Vera) and S-form verapamil enantiomer (S-Vera) were evaluated in mouse models and R-Vera seems to represent an effective option in diabetes treatment by downregulating TXNIP and reducing ß-cell apoptosis with an established safety profile and only weak adverse cardiac effects, such as negative inotropy. While the rise of intracellular calcium concentration is known in general as the main trigger of exocytosis and subsequent insulin secretion, verapamil reduces by blocking calcium channels the intracellular calcium concentration and prevents long term ß-cell impairment which is partly caused by chronic increased intracellular Ca2+ levels. This preventive mechanism contributes to preserved ß-cell function by TXNIP downregulation, ameliorating less apoptosis in pancreatic ß-cells and helps to preserve continuously endogenous insulin levels during glucose metabolism regulation.”
Reviewer 2 Report
Comments to Author
Title: Verapamil and its role in diabetes
Although the study looks interesting, there are some minor corrections with the following findings:
1. English language needs to be improved throughout the manuscript.
2. Introduction: The authors have written this section quite comprehensively; however, I would consider writing more descriptively on the molecular mechanism of insulin secretion from pancreatic beta cells.
3. I would suggest to add a method section in the manuscript by emphasizing how did author collect the information including key points.
4. The author's explanation for their findings appears to be promising. However, I would suggest excluding their own future plans and their present ongoing research proposal from the discussion. But I would recommend including the limitation and future prospective of these findings
5. Overall, the findings look interesting,
Author Response
Dear Editor,
Dear Reviewers,
Thank you very much for reviewing our manuscript entitled “Verapamil and its role in diabetes”. Please find below a point-to-point response to the specific comments.
Comments to Author
Title: Verapamil and its role in diabetes
Although the study looks interesting, there are some minor corrections with the following findings:
We highly appreciate the willingness to review our manuscript and also express our thanks for the comments and suggestions for the authors. Please find below a point-to-point response to the specific comments.
- English language needs to be improved throughout the manuscript.
We would like to thank the reviewer for addressing this important point to review the grammar as well as English formatting and thank for the suggestions from the reviewer. We are aware about this inaccuracy and revised our manuscript by an English native and included several new text paragraphs.
- Introduction: The authors have written this section quite comprehensively; however, I would consider writing more descriptively on the molecular mechanism of insulin secretion from pancreatic beta cells.
Thank you very much for this important comment. In our revised manuscript we included the molecular mechanism of insulin secretion in the introduction part and the revised our manuscript, which was amended as following, on page 2, line 49 to 61:
“The primary physiological stimulus for insulin secretion is known to be the increase of circulating glucose concentration. The direct insulin secretion by glucose involves a “triggering” and an “amplifying” pathway. The “triggering” pathway is activated by several biochemical signals, involving the adenosin triphosphat (ATP) generation by glucose metabolism, the closure of ATP-sensitive potassium (K ATP) channels resulting in membrane depolarization and consequent activation of voltage-gated calcium channels. The subsequent sharp rise of intracellular calcium levels contributes to the triggered exocytosis of readily releasable pooled insulin secretory granules by membrane fusion and release to the cell exterior. After the “first phase” of insulin release resulting in a sharp peak, the amplifying pathway provides lower but sustained insulin release for several hours in the “second phase” of insulin secretion. The amplifying pathway is activated in the presence of maximal intracellular Ca2+ levels and is largely independent of K ATP driven mechanisms.”
- I would suggest to add a method section in the manuscript by emphasizing how did author collect the information including key points.
Thank you very much for this important comment. In our revised manuscript we included a method section to emphasize the collection of relevant information on the topic and revised the manuscript as following, on page 4, line 177 to 186:
“2. Material and Methods
2.1 Scientific research
We selected relevant scientific research published from October 1984 until May 2022 by searching PubMed. Potentially eligible studies were considered to be included in our narrative review after searching by combined term medial subject headings and keywords, such as type 1 diabetes (T1D), type 2 diabetes (T2D), insulin secretion, ß-cell preservation, verapamil, and Thioredoxin-interacting protein (TXNIP). After completing the search 69 papers and one web source were included to detail the systemic and cellular effects of orally administrated CCB verapamil in T1D and T2D subjects.”
- The author's explanation for their findings appears to be promising. However, I would suggest excluding their own future plans and their present ongoing research proposal from the discussion. But I would recommend including the limitation and future prospective of these findings.
Thank you very much for the comment on the organization of the discussion section. We appreciate the reviewer´s suggestions to improve the structure and quality of our work. In the revised version of our manuscript these major comments were processed, the detailed information of both studies was deleted and our discussion section was amended as following on page 15, line 743 to 750 and page 15, line 763 to page 16, line 772:
“In this context, the VER-A-T1D trial (VER-A-T1D; NCT04545151) as a multicenter, randomized, double blind, placebo-controlled study will evaluate the effect of orally administered verapamil on the preservation of ß-cell function as measured by stimulated c-peptide levels after 12 months. Furthermore, another multinational trial investigates the use of verapamil in children and adolescents with newly diagnosed T1D to assess hybrid closed loop therapy and verapamil for ß-cell preservation in new onset T1D (CLVer; NCT04233034), which was initiated in July 2020 and will be completed in September 2022.”
and
“Nevertheless the outcomes of both initiated studies and the presented scientific research in general are limited to a small number of participants and a short follow-up time with lack of long term follow-up results. Future research and longer follow up periods will be of great interest for the scientific community, such as safety profile and side effects, as well as their daily practicability regarding the regular continuous verapamil use as new innovative therapeutical approach. In the end the previously described positive effects of oral adjunct verapamil administration in subjects with T1D have to be confirmed in larger studies.“
- Overall, the findings look interesting.
We want to thank the reviewer for assessing our paper and we highly appreciate this positive comment. Based on the recommendation of the reviewer, our paper was modified to improve the quality of our work. Thank you very much for your support.